# A Probabilistic Model of Social Decision Making based on Reward Maximization

**Koosha Khalvati**
Department of Computer Science
University of Washington
Seattle, WA 98105
koosha@cs.washington.edu

**Seongmin A. Park**
CNRS UMR 5229
Institut des Sciences Cognitives Marc Jeannerod
Lyon, France
park@isc.cnrs.fr

**Jean-Claude Dreher**
CNRS UMR 5229
Institut des Sciences Cognitives Marc Jeannerod
Lyon, France
dreher@isc.cnrs.fr

**Rajesh P. N. Rao**
Department of Computer Science
University of Washington
Seattle, WA 98195
rao@cs.washington.edu

## Abstract

A fundamental problem in cognitive neuroscience is how humans make decisions, act, and behave in relation to other humans. Here we adopt the hypothesis that when we are in an interactive social setting, our brains perform Bayesian inference of the intentions and cooperativeness of others using probabilistic representations. We employ the framework of partially observable Markov decision processes (POMDPs) to model human decision making in a social context, focusing specifically on the volunteer's dilemma in a version of the classic Public Goods Game. We show that the POMDP model explains both the behavior of subjects as well as neural activity recorded using fMRI during the game. The decisions of subjects can be modeled across all trials using two interpretable parameters. Furthermore, the expected reward predicted by the model for each subject was correlated with the activation of brain areas related to reward expectation in social interactions. Our results suggest a probabilistic basis for human social decision making within the framework of expected reward maximization.

## 1 Introduction

A long tradition of research in social psychology recognizes volunteering as the hallmark of human altruistic action, aimed at improving the survival of a group of individuals living together [15]. Volunteering entails a dilemma wherein the optimal decision maximizing an individual's utility differs from the strategy which maximizes benefits to the group to which the individual belongs. The "volunteer's dilemma" characterizes everyday group decision-making whereby one or few volunteers are enough to bring common goods to the group [1, 6]. Examples of such volunteering include vigilance duty, serving on school boards or town councils, and donating blood. The fact that makes the volunteer's dilemma challenging is not only that a lack of enough volunteers would lead to no common goods being produced, but also that resources would be wasted if more than the required number of group members volunteer. As a result, to achieve maximum utility in the volunteer's dilemma, each member must have a very good sense of others' intentions in the absence of any

---

This work was supported by LABEX ANR-11-LABEX-0042, ANR-11-IDEX-0007, NSF-ANR 'Social_POMDP' and ANR BrainCHOICE n°14-CE13-0006 to JC. D, NSF grants EEC-1028725 and 1318733, ONR grant N000141310817, and CRCNS/NIMH grant 1R01MH112166-01.

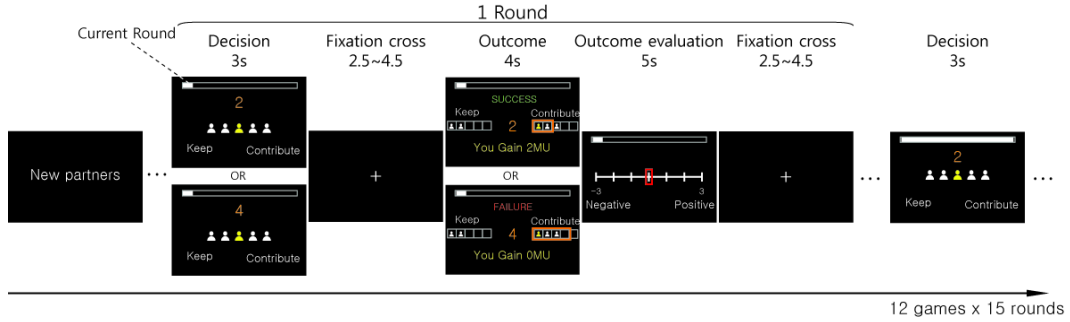

Figure 1: The computer screen that players see during one round of PGG.

communication between them, before choosing their actions. To model such social decision making, one therefore needs a framework that can model the uncertainty associated with the "theory of mind" of each player. In this paper, we tackle this problem by combining a probabilistic model with behavioral measures and fMRI recordings to provide an account of the decisions made under the volunteer's dilemma and the underlying neural computations.

The Public Goods Game (PGG) is a classic paradigm in behavioral economics. It has previously been employed as a useful tool to study the neural mechanisms underlying group cooperation [2, 3, 4, 16]. Here we recast the PGG to investigate the volunteer's dilemma and conducted an experiment where 30 subjects played a version of the PGG while their brain activity was simultaneously recorded using fMRI. We show how a probabilistic model based on Partially Observable Markov Decision Processes (POMDPs) with a simple and intuitive state model can explain our subjects' behavior. The normative model explains the behavior of the subjects in all trials of the game, including the first trial of each round, using only two parameters. The validity of the model is demonstrated by the correlation between the reward predicted by the model and activation of brain areas implicated in reward expectation in social interactions. Also, the values of the parameters of our model are interpretable, and the differences among players and games can be explained by these parameters.

## 2 Public Goods Game and Experimental Paradigm

In a Public Goods Game (PGG), $N$ strangers make collective decisions together as a group. In the current study, we keep the number of members in a group constant at $N = 5$. No communication among members is allowed. The game is composed of 15 rounds of interactions with the same partners. At the beginning of each round, 1 monetary unit (MU) is endowed (E) to each of $N = 5$ individuals. Each individual can choose between two decisions, *contribution (c)* or *free-riding (f)*. When the participant makes a decision, the selected option is highlighted on the screen. The participant must make a decision within three seconds, otherwise a warning message appears and the trial is repeated. After all members of the group have made their decisions, the feedback screen is shown to all. According to the decisions of group members, public good is produced as the group reward ($R = 2MU$) only if at least $k$ individuals have contributed their resources ($k = 2$ or $k = 4$). Value of $k$ was conveyed to group members before decision-making and is kept fixed for any single PGG. From the feedback screen, participants only know the number of other contributors and not individual single member decisions, which are represented by white icons. A yellow icon stands for the individual playing in the scanner and served to track their decisions. Each PGG consists of a finite round of interactions ($T = 15$). This is informed to all participants. The computer screen in front of each player during one round of a PGG is shown in Figure 1. Each contribution has a cost ($C = 1MU$). Therefore, the resultant MUs after one round is $E - C + R = 2MU$ for the contributor and $E + R = 3MU$ for the free-rider when public good is produced (*SUCCESS*). On the other hand, the contributor has $E - C = 0MU$ and the free-rider has $E = 1MU$ when no public good is produced (*FAILURE*) . Each participant plays 14 games. During the first 2 PGGs, they receive no feedback, but the following 12 PGGs provide social and monetary feedback as shown in Figure 1. Our analyses are from the 12 PGGs with feedback. Importantly, we inform participants before the experiment that they get a final monetary reward as much as the result of one PGG randomly selected by the compute r at the end of the study [23].

We recruited 30 right-handed subjects to participate in the Public Goods Game and make decisions in an fMRI scanner. Data from 29 participants (fourteen women, mean age 22.97 years old 1.99 S.D.) were analyzed (one participant aborted the experiment due to anxiety). Based on self-reported questionnaires, none of our subjects had a history of neurological or psychiatric disorders. Each participant was told that they would play with 19 other participants located in another room; in actuality, a computer selected the actions instead of 19 others. Each action selected by our computer algorithm in any round is a probabilistic function of the participant's action in the previous round ($a_i^{t-1}$), and its own previous actions ($\sum_{j \in -i} a_j^{t-1}$). Given the average contribution rate of others $\bar{a}_{-i}^t = \frac{\sum_{j \in -i} a_j^t}{N-1}$ we have $logit(\bar{a}_{-i}^t) = e_0 a_i^{t-1} + e_1((\frac{1-K^{T-t+1}}{1-K})^{e_2} \bar{a}_{-i}^{t-1} - K)$ where $K = k/N$. This model has 3 free parameters: $e_0, e_1, e_2$. These are obtained by fitting the above function to the actual behavior of individuals in another PGG study [16]. Therefore, this function is a simulation of real individuals' behavior in a PGG. For the first round, we use the mean contribution rate of each subject as their fellow members' decision.

## 3  Markov Decision Processes and POMDPs

The family of Markov Decision Processes (MDPs and POMDPs) provide a mathematical framework for decision making in stochastic environments [22]. A Markov Decision Process (MDP) is formally a tuple $(S, A, T, R, \gamma)$ with the following description: $S$ is the set of states of the environment, $A$ is the set of actions, $T$ is the transition function $P(s|s', a)$, i.e., the probability of going from a state $s'$ to state $s$ after performing action $a$. $R : S \times A \to \mathbb{R}$ is a bounded function determining the reward obtained after performing action $a$ in state $s$. $\gamma$ is the discount factor which we assume here is 1. Starting from an initial state $s_0$, the goal is to find the sequence of actions to maximize expected discounted reward $E_{s_t}[\sum_{t=0}^{\infty} \gamma^t R(s_t, a_t)]$. This sequence of actions is given by an optimal policy, which is a mapping from states to actions: $\pi^* : |S| \to |A|$ representing the best action at a given state. The optimal policy can be computed by an efficient algorithm called *value iteration* [22]. MDPs assume that the current state is always fully observable to the agent. When this is not the case, a more general framework, known as Partially Observable Decision Processes (POMDPs), can be used. In a POMDP, the agent reasons about the current state based on an *observation*. Therefore, POMDP can be regarded as an MDP with observations, $Z$ and an observation function $O : Z \times A \times S \to [0, 1]$, which determines $P(z|a, s)$, the probability of observing $z$ after performing action $a$ in state $s$. In a POMDP, instead of knowing the current state, $s_t$, the agent computes the belief state, $b_t$, which is the posterior probability over states given all past observations and actions. The belief state can be updated as: $b_{t+1}(s) \propto O(s, a_t, z_{t+1}) \sum_{s'} T(s', s, a_t) b_t(s')$. Consequently, the optimal policy of a POMDP is a mapping from belief state to actions: $\pi^* : B \to A$ where $B = [0, 1]^{|S|}$. One could easily see that a POMDP is an MDP whose states are belief states. As the belief state space is exponentially larger than the original state space ($B = [0, 1]^{|S|}$), solving POMDPs is computationally very expensive (*NP-hard* [19]). Therefore, heuristic methods are used to approximate the optimal policy for a POMDP [11, 20]. In the case that the belief state can be expressed in closed form, e.g., Gaussian, one can solve the POMDP by considering it as an MDP whose state space is the POMDP's belief state space and performing the value iteration algorithm. We use this technique in our model.

## 4  Model of the Game

In a PGG with $N$ players and known minimum number of required volunteers ($k$), the reward of a player, say player $i$, in each round is determined only by their action (free ride ($f$) or contribution ($c$)), and the total number of contributors among other players. We use the notation $-i$ to represent all players except player $i$. We denote the action of each player as $a$ and the reward of each player as $r$. The occurrence of an event is given by an indicator function $\mathbb{I}$ (for event $x$, $\mathbb{I}(x)$ is equal to 1 if event $x$ happens and 0 otherwise). Then, the reward expected by player $i$ at round $t$ is:

$$
\begin{aligned}
r_t^i &= \mathbf{E}\left[ E - \mathbb{I}(a_t^i = c) \cdot C + \mathbb{I}\left( \sum_{j=1}^{N} \mathbb{I}(a_t^j = c) \geq k \right) \cdot R \right] \\
&= \mathbf{E}\left[ E - \mathbb{I}(a_t^i = c) \cdot C + \mathbb{I}\left( \mathbb{I}(a_t^i = c) + \sum_{j \in -i} \mathbb{I}(a_t^j = c) \geq k \right) \cdot R \right]
\end{aligned}
\tag{1}
$$

This means that in order to choose the best action at step $t$ $(a_t^i)$, player $i$ should estimate the probability of $\sum_{j \in -i} \mathbb{I}(a_t^j = c)$. Now if each player is a contributor with probability $\theta_c$, the probability of this sum would be a binomial distribution:

$$P(\sum_{j \in -i} \mathbb{I}(a_t^j = c) = k') = \binom{N-1}{k'} \theta_c^{k'} (1 - \theta_c)^{N-1-k'} \tag{2}$$

We could model the whole group with one parameter, because players only get the total number of contributions made by others and not individual contributions. Individuals cannot be tracked by others, and all group members can be seen together as one group. In other words, $\theta_c$ could be interpreted as cooperativeness of the group on average. With $\theta_c$, the reward that player $i$ expects at time step $t$ is:

$$r_t^i = E - \mathbb{I}(a_t^i = c) \cdot C + \mathbb{I}(a_t^i = c). \left( \sum_{k'=k-1}^{N-1} \binom{N-1}{k'} \theta_c^{k'} (1 - \theta_c)^{N-1-k'} \right) \cdot R$$
$$+ \mathbb{I}(a_t^i = f). \left( \sum_{k'=k}^{N-1} \binom{N-1}{k'} \theta_c^{k'} (1 - \theta_c)^{N-1-k'} \right) \cdot R \tag{3}$$

This is only for one round. The game however, contains multiple rounds (15 here) and the goal is to maximize the total expected reward, not the reward of a specific round. In addition, $\theta_c$ changes after each round because players see others' actions and update the probability of cooperativeness in the group. For example, if a player sees that others are not contributing, they may reduce $\theta_c$ when picking an action in the next round. Also, since our subjects think they are playing with other humans, they may assume others make these updates too. As a result, each player thinks their own action will change $\theta_c$ as well. In fact, although they are playing with computers, our algorithm does depend on their actions and their assumption is thus, correct. In addition, because subjects think they have a correct model of the group, they assume all group members have the same $\theta_c$ as them. If we define each possible value of $\theta_c$ as a discrete state (this set is infinite, but we could discretize the space, e.g., 100 values from 0 to 1) and model the change in $\theta_c$ with a transition function, our problem of maximizing total expected reward becomes equivalent to an MDP.

Unfortunately, the subject does not know $\theta_c$ and therefore must maintain a probability distribution (belief state) over $\theta_c$ denoting *belief* about the average cooperativeness of the group. The model therefore becomes a POMDP. The beta distribution is a conjugate prior for the binomial distribution, meaning that when the prior distribution is a beta distribution and the likelihood function is a binomial distribution, the posterior will also be a beta distribution. Therefore, in our model, the subject starts with a beta distribution as their initial belief, and updates their belief over the course of the game using the transition and observation functions which are both binomial, implying that their belief always remains a beta distribution. The beta distribution contains two parameters, $\alpha$ and $\beta$. Using maximum likelihood estimation (MLE), the posterior distribution after seeing $k'$ true events from total of $N$ events with prior $Beta(\alpha, \beta)$ is $Beta(\alpha + k', \beta + N - k')$:

$$\text{Prior}: Beta(\alpha, \beta) \rightarrow P(\theta) = \frac{\theta^{\alpha-1}(1-\theta)^{\beta-1}}{B(\alpha, \beta)} \tag{4}$$

$$\text{Posterior}: Beta(\alpha + k', \beta(t) + N - k') \rightarrow P(\theta) = \frac{\theta^{\alpha+k'-1}(1-\theta)^{\beta+N-k'-1}}{B(\alpha + k', \beta + N - k')} \tag{5}$$

where $B(\alpha, \beta)$ is the normalizing constant: $B(\alpha, \beta) = \int_0^1 \theta^{\alpha-1}(1-\theta)^{\beta-1}d\theta$.

As mentioned before, each POMDP is an MDP whose state space is the belief state of the original POMDP. As our belief state has a closed form, we can estimate the solution of our POMDP by discretizing this belief space, e.g., considering a bounded set of integers for $\alpha$ and $\beta$, and solving it as an MDP. Also, the transition function of this MDP would be based on the maximum likelihood estimate shown above. This transition function is as follows:

$$P((\alpha + k' + 1, \beta + N - 1 - k')|(\alpha, \beta), c) = \binom{N-1}{k'} \frac{B(\alpha + k', \beta + N - 1 - k')}{B(\alpha, \beta)}$$
$$P((\alpha + k', \beta + N - k')|(\alpha, \beta), f) = \binom{N-1}{k'} \frac{B(\alpha + k', \beta + N - 1 - k')}{B(\alpha, \beta)} \tag{6}$$

The pair $(\alpha, \beta)$ is the state and represents the belief of the player about $\theta_c$, given by $Beta(\alpha, \beta)$. The reward function of this belief-based MDP is:

$$R((\alpha, \beta), c) = E - C + \sum_{k'=k-1}^{N} \binom{N-1}{k'} \frac{B(\alpha + k', \beta + N - 1 - k')}{B(\alpha, \beta)} R$$

$$R((\alpha, \beta), f) = E + \sum_{k'=k}^{N} \binom{N-1}{k'} \frac{B(\alpha + k', \beta + N - 1 - k')}{B(\alpha, \beta)} R$$

(7)

This MDP shows how the subject plays and learns their group dynamics simultaneously by updating their belief about the group during the course of the game. Note that although we are reducing the problem to an MDP for computational efficiency, conceptually the player is being modeled by a POMDP because the player maintains a belief about the environment and updates it based on observations (here, other players' actions).

## 5 Results

The parameters of our model are all known, so the question is how the model differs for different individuals. The difference is in the initial belief of the player about the group that they are playing within, in other words, the state that our belief-based MDP starts from ($b_0$ in POMDP parlance). This means that each individual has a pair $\alpha_0$ and $\beta_0$ for each $k$ that shapes their behavior through the game. For example, an $\alpha_0$ significantly larger than $\beta_0$ means that the player starts the game with the belief that the group is cooperative. Also, $\alpha_0$ and $\beta_0$ for the same individual differs for different $k$'s since the number of required volunteers changes the game and consequently the belief of the player about the optimal strategy. We investigate these differences in our analysis below.

### 5.1 Modeling behavioral data

To find $\alpha_0$ and $\beta_0$ of each player (and for each $k$), we run our model with different values of $\alpha_0$ and $\beta_0$, and using the actions that the player sees as other players' actions during the experiment, we check if the actions predicted by our model is the same as the actual actions of the player. In other words we find the $\alpha_0$ and $\beta_0$ that minimize $\sum_{t=1}^{15} |a_t^i - \tilde{a}_t^i|$ where $a_t^i$ is the action of our subject at step $t$, and $\tilde{a}_t^i$ is the predicted action from our model. Note that we only give the model other players' data and do not correct the predicted action for the previous state if the model has made a mistake. Also, we calculate the error on all games of that player with the same $k$, i.e., we assume each game is an independent data point. This is justified because subjects are explicitly told that at each game they could play with different players and also, they get reward for only one game chosen randomly. As a result one cannot use one game for training for the next ones. For each player, we call the average of $\sum_{t=1}^{15} |a_t^i - \tilde{a}_t^i|$ among all of their games with the same $k$, *round by round error*.

The average round by round error among all players for the POMDP was 3.38 for $k = 2$ and 2.15 for $k = 4$ (Table 1). For example, only around 2 out of 15 rounds were predicted incorrectly by our model for $k = 4$. The possible $\alpha_0$ and $\beta_0$ values for each player ranged over all integers between 1 and 100, yielding $100^2$ pairs to evaluate as $s_0$ for the belief-based MDP; this evaluation process was computationally efficient. We found that MDPs with horizons longer than the true number of rounds fit our data better. As a result, we set our horizon to a number much larger than 15, in this case, 50. Such an error in estimating the dynamics of a game in humans is consistent with previous reports [3].

To compare our results with other state-of-the-art methods, we fit a previously proposed descriptive model [24] to our data. This model assumes that the action of the player in each round is a function of their action in the previous round (the "Previous Action" model). Therefore, to fit the data we need to estimate $p(a_t^i | a_{t-1}^i)$. This means that the model has two parameters, i.e. $p(a_t^i = c | a_{t-1}^i = c)$ and $p(a_t^i = c | a_{t-1}^i = f)$. Note that this descriptive model is unable to predict the first action. We found that its average round by round error for the last 14 rounds (3.90 for $k = 2$ and 3.25 for $k = 4$) is more than the POMDP model's error (Table 1), even though it considers one round less than the POMDP model.

We also used Leave One Out Cross Validation (LOOCV) to compare the models (see Table 1). Although the LOOCV error for $k = 2$ is larger for the POMDP model, the POMDP model's error

Table 1: Average round by round error by POMDP, the descriptive model based on previous action, $p(a_t^i | a_{t-1}^i)$, and the most general descriptive mode, $p(a_t^i | a_{t-1}^i, \sum_{j \in -i} a_{t-1}^j)$ . In front of each error, the normalized error (divided by number of rounds) is written in parenthesis to facilitate comparison.

| model | Fitting error $(k=2)$ | Fitting error $(k=4)$ | LOOCV error $(k=2)$ | LOOCV error $(k=4)$ | Total number of rounds |
|---|---|---|---|---|---|
| POMDP | 3.38 (**0.22**) | 2.15 (**0.14**) | 4.23 (**0.28**) | 2.67 (**0.18**) | 15 |
| Previous Action | 3.90 (0.28) | 3.25 (0.23) | 4.00 (0.29) | 3.48 (0.25) | 14 |
| All Actions | 3.75 (0.27) | 2.74 (0.19) | 5.52 (0.39) | 7.33 (0.52) | 14 |

is for 15 rounds while the error for the descriptive model is for 14 (note that the error divided by number of rounds is larger for the descriptive model). In addition, to examine if another descriptive model based on previous rounds can outperform the POMDP model, we tested a model based on all previous actions, i.e. $p(a_t^i | a_{t-1}^i, \sum_{j \in -i} a_{t-1}^j)$. The POMDP model outperforms this model as well.

## 5.2 Comparing model predictions to neural data

Besides modeling human behavior better than the descriptive models, the POMDP model can also predict the amount of reward the subject is expecting since it is formulated based on reward maximization. We use the parameters obtained by the behavioral fit and generate the expected reward for each subject before playing the next round.

To validate these predictions about reward expectation, we checked if there is any correlation between neural activity recorded by fMRI and the model's predictions. Image preprocessing was performed using the SPM8 software package. The time-series of images were registered in three-dimensional space to minimize any effects from the participant's head motion. Functional scans were realigned to the last volume, corrected for slice timing, co-registered with structural maps, spatially normalized into the standard Montreal Neurological Institute (MNI) atlas space, and then spatially smoothed with an $8mm$ isotropic full-width-at-half-maximum (FWHM) Gaussian kernel using standard procedures in SPM8. Specifically, we construct a general linear model (GLM) and run a first-level analysis modeling brain responses related to outcome while informing the judgments of others. They are modeled as a box-car function time-locked to the onset of outcome with the duration of 4 sec. Brain responses related to decision-making with knowledge of the outcome of the previous trial are modeled separately. These are modeled as a box-car function time-locked to the onset of decision-making with duration of reaction times in each trial. They are further modulated by parametric regressors accounting for the expected reward. In addition, the six types of motion parameters produced for head movement, and the two motor parameters produced for buttons pressing with the right and the left hands are also entered as additional regressors of no interest to account for motion-related artifacts. All these regressors are convolved with the canonical hemodynamic response function. Contrast images are calculated and entered into a second-level group analysis. In the GLM, brain regions whose blood-oxygen-level-dependent (BOLD) response are correlated with POMDP-model-based estimates of expected reward are first identified. To correct for multiple comparisons, small volume correction (SVC) is applied to a priori anatomically defined regions of interests (ROI). The search volume is defined by a $10mm$ diameter spherical ROI centered on the dorsolateral prefrontal cortex (dlPFC) and the ventral striatum (vS) that have been identified in previous studies. The role of the dlPFC has been demonstrated in the control of strategic decision-making [14], and its function and gray matter volume have been implicated in individual differences in social value computation [7, 21]. Moreover, vS has been found to mediate rewards signal engaged in mutual contribution, altruism, and social approval [18]. In particular, the left vS has been found to be associated with both social and monetary reward prediction error [13].

We find a strong correlation between our model's prediction of expected reward and activity in bilateral dlPFC (the peak voxel in the right dlPFC: $(x, y, z) = (42, 47, 19)$, $T = 3.45$, and the peak voxel in the left dlPFC: $(x, y, z) = (-30, 50, 25)$, $T = 3.17$) [7], and left vS (the peak voxel in the vS: $(x, y, z) = (-24, 17, -2)$, $T = 2.98$) [13](Figure 2). No other brain area was found to have a higher activation than them at the relatively liberal threshold, uncorrected $p < 0.005$. Large activations were found in these regions when participants received the outcome of a trial ($p < 0.05$, FWE corrected within small-volume clusters). This is because after seeing the outcome of one round, they update their belief and consequently their expected reward for the next round.

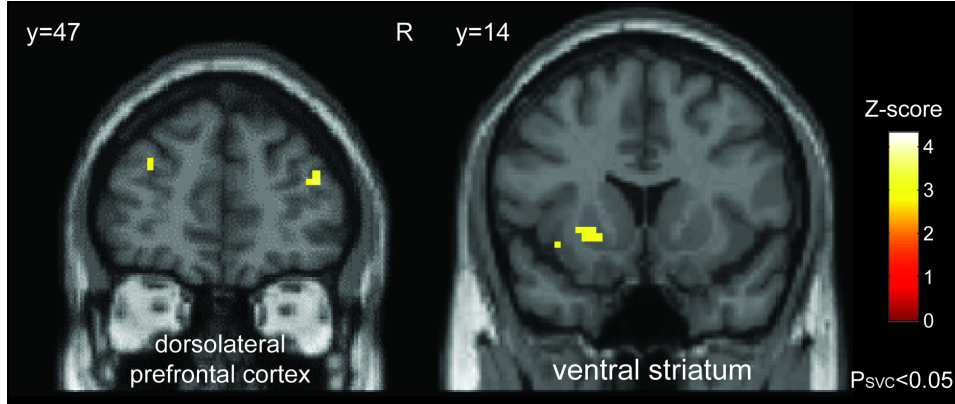

Figure 2: Strong correlation between brain activity in the dlPFC and the left vS after seeing the outcome of a round and the predicted expected reward for the next round by our model. The activations were reported with a significance of $p < 0.05$, FWE across participants corrected in a priori region of interest. The activation maps are acquired at the threshold, $p < 0.005$ (uncorrected). The color in each cluster indicates the level of z-score activation in each voxel.

## 5.3 Modeling subjects' perception of group cooperativeness

The ratio and the sum of the best fitting $\alpha_0$ and $\beta_0$ that we obtain from the model are interpretable within the context of cognitive science. In the Beta-binomial distribution update equations 4 and 5, $\alpha$ is related to the occurrence of the action "contribution" and $\beta$ to "free-riding." Therefore, the ratio of $\alpha_0$ to $\beta_0$ captures the player's prior belief about the cooperativeness of the group. On the other hand, after every binomial observation (here round), $N$ (here $N = 5$) is added to the prior. Therefore, the absolute values of $\alpha$ and $\beta$ determine the weight that the player gives to the prior compared to their observations during the game. For example, adding 5 does not change $Beta(100, 100)$ much but changes $Beta(2, 2)$ to $Beta(7, 2)$; the former does not alter the chance of contribution versus free riding much while the latter indicates that the group is cooperative.

We estimated the best initial parameters for each player, but is there a unique pair of $\alpha_0$ and $\beta_0$ values that minimizes the round by round error or are there multiple values for the best fit? We investigated this question by examining the error for all possible parameter values for all players in our experiments. The error, as a function of $\alpha_0$ and $\beta_0$, for one of the players is shown in Figure 3a as a heat map (darker means smaller error, i.e. better fit). We found that the error function is continuous and although there exist multiple best-fitting parameter values, these values define a set of lines $\alpha = a\beta + c$ with bounds $min \leq \alpha \leq max$. The lines and bounds are linked to the ratio and prior weight alluded to above, suggesting that players do consider prior probability and the weight, and best-fitting $\alpha_0$ and $\beta_0$ values have similar characteristics.

We also calculated the average error function over all players for both values of $k$. As shown in figures 3b and 3c, $\alpha_0$ is larger than $\beta_0$ for $k = 2$, while for $k = 4$, they are close to each other. Also, the absolute value of these parameters are larger for $k = 2$. This implies that when $k = 4$, players start out with more caution to ascertain whether the group is cooperative or not. For $k = 2$ however, because only 2 volunteers are enough, they start by giving cooperativeness a higher probability. Higher absolute value for $k = 2$ is indicative of the fact that the game tends towards mostly free riders for $k = 2$ and the prior is weighted much more than observations. Players know only 2 volunteers are enough and they can free-ride more frequently but still get the public good.[1]

## 6 Related Work

PGG has previously been analyzed using descriptive models, assuming that only the actions of players in the previous trial affect decisions in the current trial [8, 24, 25]. As a result, the first trial of each round cannot be predicted by these models. Moreover, these models only predicts with what probability each player changes their action. The POMDP model, in contrast, takes all trials of each

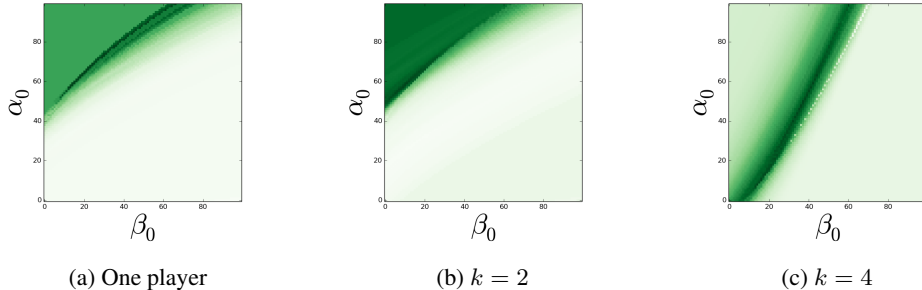

(a) One player          (b) $k = 2$          (c) $k = 4$

Figure 3: Round by round error based on different initial parameters $\alpha_0$ and $\beta_0$. Darker means lower error. (a) Error function for one of the players. The function for other players and other $k$s has the same linear pattern in terms of continuity but the location of the low error line is different among individuals and $k$s. (b) Average error function over all players for $k = 2$. (c) Average error function for $k = 4$.

round into account and predicts actions based on prior belief of the player about the cooperativeness of the group, within the context of maximizing expected reward. Most importantly, the POMDP model predicts not only actions, but also the expected reward for the next round for each player as demonstrated in our results above.

POMDPs have previously been used in perceptual decision making [12, 17] and value-based decision making [5]. The modeled tasks, however, are all single player tasks. A model based on iPOMCPs, an interactive framework based on POMDPs ([9]) with Monte Carlo sampling, has been used to model a trust game [10] involving two players. The PGG task we consider involves a larger group of players (5 in our experiments). Also, the iPOMCP algorithm is complicated and its neural implementation remains unclear. By comparison, our POMDP model is relatively simple and only uses two parameters to represent the belief state.

## 7 Discussion

This paper presents a probabilistic model of social decision making that not only explains human behavior in volunteer's dilemma but also predicts the expected reward in each round of the game. This prediction was validated using neural data recorded from an fMRI scanner. Unlike other existing models for this task, our model is based on the principle of reward maximization and Bayesian inference, and does not rely on a subject's actions directly. In other words, our model is normative. In addition, as we discussed above, the model parameters that we fit to an individual or $k$ are interpretable.

One may argue that our model ignores empathy among group members since it assumes that the players attempt to maximize their own reward. First, an extensive study with auxiliary tasks has shown that pro-social preferences such as empathy do not explain human behaviour in the public goods games [3]. Second, one's own reward is not easily separable from others' rewards as maximizing expected reward requires cooperation among group members. Third, a major advantage of a normative model is the fact that different hypotheses can be tested by varying the components of the model. Here we presented the most general model to avoid over-fitting. Testing different reward functions could be a fruitful direction of future research.

Although we have not demonstrated that our model can be neurally implemented in the brain, the model does capture the fundamental components of social decision making required to solve tasks such as the volunteer's dilemma, namely, belief about others (belief state in our model), updating of belief with new observations, knowing that other group members will update their beliefs as well (modeled via transition function), prior belief about people playing the game (ratio of $\alpha_0$ to $\beta_0$), weight of the prior in comparison to observations (absolute value of initial parameters), and maximizing total expected reward (modeled via reward function in MDP/POMDP). Some of these components may be simplified or combined in a neural implementation but we believe acknowledging them explicitly in our models will help pave the way for a deeper understanding of the neural mechanisms underlying human social interactions.

## Footnotes

[1]We should emphasize that this is the behavior on average and a few subjects do deviate from this behavior.

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
