[Reviews · NeurIPS 2016]

Reviewer 1

Summary

In this paper authors propose the POMDP model to simulate human decision making in the Public Goods Game (PGG). They explain the machinery of the model, its fit to behavioral PGG data and comparison to other similar models. They also suggest that the model has neural correlates with reward expectations correlated with activity in two brain areas.

Qualitative Assessment

This is an interesting modeling and model-based analysis study, providing insights into the machinery of human social decision making and possibly its neural correlates. The paper is generally well written and combines advances that could be interesting to both experimental and modeling audience. However, some of its aspects (particularly interpretation of estimated parameters and fMRI analyses) should be improved for it to be acceptable to NIPS. More specific comments: - What is the meaning of groups of 5 individuals if computer generates actions of 19 others? More details about how these actions are generated would be helpful. - Biological plausibility of the used value iteration algorithm should be discussed. - Equation 1 is hard to understand: what is the meaning of the second E on each line? - Why is it assumed that all group members have the same theta_c as the subject? - Regarding table 1: the models should be better explained. What are the free parameters/their number for each model? That should normally be included in model comparison. Which differences are statistically significant? - fMRI small volume correction (SVC) based on 10mm (radius or diameter?) spheres should be justified based on literature. How is the region center chosen for this purpose? - There is a typo in the first T value - it should be 3.45 I suppose - T values are too small to be called "strong correlation". Generally, whole brain analyses are preferable to SVC. - At the beginning of 5.3 do you mean "the best FITTING alpha_0 and beta_0"? - Do alpha and beta themselves have no neural correlates? Either way this should be reported. Alpha and beta have nice mathematical grounding but their suitability for psychological interpretation is less clear. It seems ratios or sums could be more informative. --- Response to author rebuttal: Thanks for addressing some of my concerns. However, I would like to clarify a few aspects which should be addressed in the final version. * While the use of SVC is understandable, it may lead to missing important effects in brain areas outside of the defined ones, whose relevance could also be justified and whose peak T-values may even be higher than of the reported ones. For example, many SVC-based studies of memory focus on hippocampus and exclude parahippocampal gyrus or prefrontal cortex, which were shown to have pronounced activity patterns in whole-brain studies. For this reason please provide whole-brain statistics as well (if necessary, in supplementary material), including peaks outside of the defined area with larger T values than reported. * If ratios and sums of alpha and beta are more informative, why not correlate them with neural data or behavior instead of simply mentioning this aspect in the discussion? If there are no effects, this should be reported as well.

Confidence in this Review

2-Confident (read it all; understood it all reasonably well)


Reviewer 2

Summary

The authors explore how people make strategic decisions in games with 5-person groups, where the payoff is structured such that the group is paid more as long as “enough” people decide to incur a small cost (known as a “Public Goods Game” in the behavioral economics literature). To do so, they model each agent in the game as a partially observable Markov decision process (POMDP), derive an efficient algorithm for deriving model results, and compare those results to the behavioral and neural (fMRI signals) results of a novel experiment. They find that their model predicts behavior well, better than versions of their model that other models, and that there was a correspondence between model predictions and a brain area known to be sensitive to perceived expected reward.

Qualitative Assessment

This is an excellent submission. It is well written and appropriate for the cognitive and neural modeling portions of the NIPS audience. Its contribution is largely empirical – the model is not new to the NIPS community and the general framework has been used several times in the past at NIPS. Regardless, I still believe that the experimental results are strong enough to warrant acceptance. My main criticism is that the experiments are still extremely artificial and I would love to see more realistic studies in this area. However, this is a critique of a broad and prominent literature and not just this particular submission. Also, it would probably be good to cite/relate the work to other computational models of social cognition that have been prominent at NIPS and use the same/similar framework (the work of Chris Baker, Josh Tenenbaum, and colleagues). Minor issues: Figure 1: resolution of text is too poor to read easily. Page 2, line 48-49: I would rename the variable E or use the fancy math E for expectation so that the two do not have the same letter. Page 3, line 79: Was it confirmed post-experiment that participants believed there were different subjects for each game? Page 3, line 81: What was the function? Page 3, Equation 1: Please improve the parentheses by either alternating brackets and parentheses more and/or \left and \right in LaTeX. Page 8, lines 297-298: It is implied (or at least I read it this way) that normative models and other-regarding preferences. In fact, there are many that do (e.g., Fehr-Schmidt).

Confidence in this Review

2-Confident (read it all; understood it all reasonably well)


Reviewer 3

Summary

This paper presents a model of social decision making that not only explains human behaviour but also predicts the reward in a round of game as the volunteer's game. The prediction was conform to neural data recorded from fMRI scanner. The originality here comes from the fact that the proposed model is normative since it is based on the principle of reward maximisation and Bayesian inference and does not rely on a subject's actions directly.

Qualitative Assessment

An interesting article, particularly from the point of view of the validation that is not usual in NIPS community. The model is based on POMDPs and solved via a technique using it as a MDP with belief state space. The prediction was conform to neural data recorded from fMRI scanner. The originality here comes from the fact that the proposed model is normative since it is based on the principle of reward maximisation and Bayesian inference (Via POMDPs) and does not rely on a subject's actions directly. Here some comments/suggestions/questions.... (1) In PGG, when we have ZERO contribution by all agents, it is a Nash Equilibrium, why you did not talk about it? (2) Why you do not talk about Game Theory? can we mix POMDPs and Game Theory in the context of PGG ? (3) in line 63, I do not see how we can obtain E=1MU if no one contributes?

Confidence in this Review

1-Less confident (might not have understood significant parts)


Reviewer 4

Summary

This paper applies a parametric Bayesian partially observable markov decision process (pomdp) to model both behavior and brain imaging activity for social decision making. The scenario studied is one of a repeated multi-agent partially observable game, in which agents can choose whether or not to pay a small cost and volunteer, and the entire group a larger reward if more than some minimum number of agents volunteers. The human subject plays as one of the multiple agents in a group with the other players controlled by computer simulation (unknown to the human), and the subject observes only the number of volunteers from the group. The experiment consists of two main tasks 1) learning a cooperative parameter of the pomdp model and using it to predict the action of the subject; and 2) affirming regions of the brain that correlate with social decision making and interactions.

Qualitative Assessment

Clear, well-written paper with nice experiment and interesting results. While the modeling using such a simple parametric model in a Bayesian pomdp is not a new method, its application to social decision making and its connection to real brain imaging data is compelling. It would be useful to see related works earlier, so that the comparisons in Table 1 are placed within the context of existing literature (this can even be explicit with citations) and the contributions of this work are more apparent.

Confidence in this Review

2-Confident (read it all; understood it all reasonably well)


Reviewer 5

Summary

The public goods game has been popular as for studying the behavioral economics and neuroeconomics of cooperation. The authors model the players as biased coins, and model gameplay using a POMDP in which the bias of the coin is the hidden variable. They convert this into an MDP over the belief state, after pointing out that all possible Bayesian beliefs can be described by a beta distribution with integral parameters. They compare the model with behavioral data and brain imaging data.

Qualitative Assessment

If the authors' summary of the prior literature is correct, the POMDP model seems new and interesting as it is a normative model, whereas previous models were descriptive.

Confidence in this Review

2-Confident (read it all; understood it all reasonably well)


Reviewer 6

Summary

The authors provide a model of value based decision making in a repeated public goods game, wherein subjects had a binary choice (contribute/freeride) in every round. The task is framed as a probe into volunteering behaviour (investing in a public good, that will only yield an outcome if enough participants commit their initial endowment). The paper presents a task model, based on a I-POMDP framework, aiming to give a more precise account than an earlier approach. fMRI data was acquired on real subjects playing this task. The model was then used to provide regressors for the analysis of the fMRI data. The results support value based decision making in a behavioural model comparison and yield plausible neural correlates.

Qualitative Assessment

The experimental methods are generally appropriate, although unlike for volunteering in general, in this experiment the payoff of volunteering is far more immediate. The restriction to binary choices prevents finer statements on willingness to cooperate in real subjects, the anonymity of other actors individual choices reduces the impact of using a group based paradigm. An issue with the analysis is the fact that the dynamics of the (simulated) partners in the group should be presented, otherwise it is hard to tell how subjects were influnced by the experimental set up. The model is an interesting case of applying POMDP methods. The authors point out the simplicity of the framework/current model, yet this appears to be more due to the task structure (binary choice, only summary partner response observable), rather than the model framework used. The reference in line 329-330 is missing the journal. Post Author Feedback: I thank the authors very much for their responses. I share concerns about selecting the specific regions of interest and think they need to be justified better and whole brain effects should be shown. It appears the model does fall into the class of I-POMDPs described in (PJ Gmytrasiewicz, P Doshi, A framework for sequential planning in multi-agent settings, Journal of Artificial Intelligence Research, 2005). The central aspect is then inference on the overall likelihood of volunteering, like learning about a single partner in an I-POMDP. In the present model the potential hierarchy of theory of mind levels/recursive model building is not used, so the present model is embedded in the I-POMDP framework as a "level 0" model. I think it is important to note this, to avoid confusion about the novel contributions the work is making (checking the source in line 281, the mentioned I-POMCP is a method rather than a framework, although the I-POMDP framework is treated there, and the model discussion following the method is about a specific task). I think it would be important to note that (I-)POMDPs as models of behaviour in social tasks have been used in fMRI before. See for instance (Xiang et al, Computational phenotyping of two-person interactions reveals differential neural response to depth-of-thought, Plos CB. 2012), which used the derived trial-to-trial expectations as fMRI regressors and is also based on parameters describing overall subject traits.

Confidence in this Review

2-Confident (read it all; understood it all reasonably well)